# Implications of MTHFD2 expression in renal cell carcinoma aggressiveness

**Rafaela V. N. Silva**[1]**, Lucas A. Berzotti**[1]**, Marcella G. Laia**[1]**, Liliane S. Araújo**[1]**, Crislaine A. Silva**[1]**, Karen B. Ribeiro**[2]**, Millena Brandão**[3]**, Adilha M. R. Michelleti**[4]**, Juliana R. Machado**[1]**, Régia C. P. Lira**[1]**\***

**1** Department of Pathology, Genetics and Evolution, Institute of Biological and Natural Sciences, Federal University of Triângulo Mineiro, Uberaba, Minas Gerais, Brazil, **2** Clinics Hospital, Federal University of Triângulo Mineiro, Uberaba, Minas Gerais, Brazil, **3** Ribeirão Preto Medical School, University of São Paulo, Ribeirão Preto, São Paulo, Brazil, **4** Department of Clinical Surgery, Federal University of Triângulo Mineiro, Uberaba, Minas Gerais, Brazil

\* regia.fusco@uftm.edu.br

**Data Availability Statement:** All relevant data are within the manuscript and its Supporting information files.

**Funding:** Supported by grant from Fundação de Amparo à Pesquisa do estado de Minas Gerais –

## Abstract

Renal cell carcinoma (RCC) is the most common type of cancer in kidney and is often diagnosed in advanced stages. Until now, there is no reliable biomarker to assess tumor prognosis during histopathological diagnosis. The Methylenetetrahydrofolate dehydrogenase 2 (MTHFD2) overexpression has been suggested as prognostic indicator for RCC, however, its protein profile needs to be clarified. This study investigated the MTHFD2 expression in different RCC cohorts, associating it with tumor characteristics and prognostic factors. Gene expression comparisons between non-neoplastic (NN) and tumor samples, as well as patients' survival analysis, were assessed using KM-Plotter tool. MTHFD2 protein pattern was evaluated in 117 RCC by immunohistochemistry and associations with prognosis, clinical and pathological data were investigated. The tumors exhibited higher *MTHFD2* transcript levels than NN, being even higher in the metastatic group. Opposite gene expression patterns were found among clear cell renal cell carcinoma (ccRCC) and pappilary renal cell carcinoma (pRCC) subtypes, showing higher and lower expressions compared to NN samples respectively. Overexpression was associated with shorter overall survival for ccRCC and pRCC subtypes, and shorter recurrence-free survival for pRCC. The immunolabeling profile varied according to tumor subtypes, with lower intensity and expression scores in ccRCC compared to pRCC and to chromophobe renal cell carcinoma (chRCC). MTHFD2 protein expression was associated with larger tumors and higher Fuhrman grades. Although prognostic value of protein immunostaining was not confirmed, patients with higher MTHFD2 tended to have lower survival rates in the pRCC group. The results highlight MTHFD2 different patterns according to RCC histological subtypes, revealing marked variations at both the genetic and protein levels. The mRNA indicated tumor prognosis, and greater expression in the tumor samples. Although MTHFD2 immunolabeling suggests tumor aggressiveness, it needs to be validated in other cohorts as potential prognostic factor.

FAPEMIG (Protocol number: APQ-00258-21) and scholarship from Coordenação de Aperfeiçoamento de Pessoal de Nível Superior (CAPES). The funders had no role in study design, data collection and analysis, decision to publish, or preparation of the manuscript.

**Competing interests:** The authors have declared that no competing interests exist.

## Introduction

Renal cell carcinoma (RCC) represents 90% of all renal tumors and about 2% of all malignancies diagnosed in adults [1]. Its incidence has increased worldwide, and the number of cases is estimated to grow significantly in countries such as Brazil, Ecuador, and Colombia [2]. In the opposite direction, the mortality rates due to RCC have declined in most countries, except for Brazil, Croatia, Greece, Ireland, and Portugal [3]. Among the three main histological variants, clear cell carcinoma (ccRCC) is the most common type (75% of cases). It is characterized by *VHL* impairment and presents the poorest prognosis with a higher frequency of metastases. The Papillary (pRCC) and chromophobe carcinomas (chRCC) represent 15% and 5% of RCC cases respectively and are less aggressive entities [1, 4–6]. The RCC usually progresses asymptomatic, and 30% of patients present metastatic disease at diagnosis [1].

In addition to the specific genetic profiles, such as genes mutations, the tumor microenvironment exerts influence on the tumor biology, playing a crucial role in the therapy response [7–12]. The RCC stands out for having one of the greatest immunological infiltrations compared to other types of cancer, highlighting its aggressive nature. The Warburg effect and activation of specific metabolic pathways are known to promote angiogenesis, inflammatory signatures and antioxidant defense, which are associated with impairment of chemotherapy and radiotherapy and malignant behavior [13–16]. In contrast, nephrectomy and targeted anti-angiogenic drugs have shown improvements in survival, being effective in patients with localized [16, 17].

Considering the RCC complexity and aggressiveness, it is urgent to apply reliable biomarkers to assess tumor prognosis, which, nowadays, is assessed using the Fuhrman system and the American Joint Committee on Cancer—AJCC staging. While the Fuhrman grade considers nuclei cell characteristics, the AJCC classifies the tumor according to the TNM system [18]. Comprehensive studies have indicated that MTHFD2 plays a significant role in cancer phenotypes, indicating consistent protein increase in various tumors, including breast, colon, liver, and kidney cancers. Moreover, MTHFD2 higher expression was associated with unfavorable prognoses in kidney cancer and pancreatic ductal adenocarcinoma [19–21].

The MTHFD2 is a mitochondrial enzyme that acts in carbon-1 folate metabolism, as well as plays an essential role in the regeneration and maintenance of NADP(H) cofactor. Generally, cancer cells present high levels of NADPH to reinforce redox defense and to increase the biosynthetic reactions that sustain their rapid growth [22]. For this reason, it is hypothesized that MTHFD2 hyperregulation contributes to cancer progression, which is related to aggressive clinicopathological parameters, metastasis, and shorter survival [19–21]. Furthermore, MTHFD2 stands out as a crucial metabolic checkpoint, that regulates both the effector and regulatory T cells, suggesting a broader role for this enzyme in RCC microenvironment context [23]. Herein, we investigated the MTHFD2 mRNA and protein expression profiles in RCC cohorts, associating them with tumor characteristics and prognostic factors.

## Material and methods

### mRNA data analysis

*MTHFD2* gene expression was analyzed according to pathological features and patient survival using the online KM-Plotter databases (https://kmplot.com/analysis/). The gene chip database [24] from the GEO (www.tnmplot.com), comprising 277 non-neoplastic renal tissue (NN), 556 tumors and 58 metastatic tumors, was investigated to identify gene expression profiles among distinct types of renal samples. Additionally, differences between NN and tumor tissues (117 NN *versus* 535 ccRCC; 77 NN *versus* 289 pRCC; 69 NN *versus* 65 chRCC), as well as

overall survival (OS) and recurrence-free survival (RFS) in each histological tumor type were evaluated in The Cancer Genome Atlas (TCGA) Pan-cancer RNAseq database [25]. For OS and RFS, the cohorts were grouped according to the median expression values by selecting the auto best cut-off option.

## Patients

This retrospective study included a total of 117 tissue fragments from patients diagnosed with RCC. Tissue specimens of surgical tumor resection were obtained from the Clinics Hospital of the Federal University of Triângulo Mineiro, and clinicopathological data was collected from patients' medical records in the period from 08/10/2022 to 12/22/2022 (S1 Appendix). To ensure complete confidentiality of the participants, we prioritized the utilization of registration numbers assigned to the paraffin blocks and slides within the surgical pathology department, in conjunction with the identifiers from the medical records, serving as the primary means of identification throughout the research.

The cohort was composed, predominantly, of male patients (65.8%), above 50 years old (86.3%), mean age of 61.8 ± 10.5 years old (range: 38–91; median = 63 years), smokers and/or alcohol consumers (59.2%) and with some comorbidity, such as diabetes mellitus, hypertension and/or chronic kidney disease (69.9%). Most cases were asymptomatic (76.1%), comprising three histological subtypes: ccRCC (77.8%), pRCC (19.9%), and chRCC (4.3%). The follow-up period ranged from 1 to 274 months. Fuhrman grade and tumor stage were carried out according to the TNM classification by American Joint Committee on Cancer (AJCC) guideline- 8th—2017 (9). All available data is summarized in S1 Table.

## Tissue microarray construction

Initially, the Haematoxylin and Eosin (HE) stained slides were reviewed by a single pathologist to confirm the diagnoses, TNM classification and Fuhrman grade. Representative areas of the tumor were marked in the slide to further obtain the tissue fragment. The tissue microarray (TMA) was constructed using two 3mm-sized cores from each case, that were then collected and reembedded in a TMA block, following protocol described previously with modifications [26]. All blocks were subsequently cut (3 μm) and the resulting slides used for immunohistochemistry technique.

## Ethical approval

The study was approved by the local Ethics Committees (protocol number: 4.981.821/2021). To guarantee patient privacy, all information used in the research was kept strictly confidential, following the ethical guidelines and privacy protocols established. The Ethics Committee waived informed consent due to the lack of intervention and the exclusion of patient-identifying information. The waiver of consent was based on three main reasons: 1- All specimens were obtained retrospectively from pathology archives. 2-There was no risk to the participants, since only anonymized tissues were used. 3- Patients' identities were anonymized and completely dissociated from any unique identifier.

## Immunohistochemistry

The reaction was performed in a dark, humid vat, using the EasyLink One System Kit (Easy-Path diagnostics, Indaiatuba, SP, Brazil), according to datasheet recommendations. First, the TMA sections were deparaffinised in xylene for 30 minutes, hydrated in decreasing concentrations of alcohol (100%, 90%, 80%, 70%), washed with water and subjected to antigen recovery

with citrate buffer (pH = 6.0) at 121˚C and 23 psi in a Pascal pan (Dako, Copenhagem, Denmark). Then, the slides were washed with distilled water and once with PBS buffer for 5 minutes. The peroxidase and protein blockers were applied for, respectively, 30 and 25 minutes at room temperature. The anti-MTHFD2 primary antibody (HPA049657, dilution 1:40, Sigma-Aldrich®, St. Louis, MO, EUA) was incubated overnight at 4˚C. Then, the slides were washed with PBS and incubated with single-step polymer for 60 minutes at room temperature. Finally, the assay followed with PBS washes, incubation with diaminobenzidine chromogen (Dako, Copenhagen, Denmark) for 5 minutes and haematoxylin counterstaining for 30 seconds.

Immunolabeling was evaluated through a semi-quantitative method based on intensity (0 = negative, 1+ = weak, 2+ = moderate, 3+ = Strong) and quantity of positive cells (0 = < 5%, 1 = 6 to 25%, 2 = 26 to 50%, 3 = 51 to 75%, 4 >76%). The score (from 0 to 12) was calculated by multiplying the intensity and the percentage of labeled cells. Two independent observers evaluated the slides blindly. For discordant duplicate samples, the most positive result was considered. Low expression was defined as a final score <4, while high expression was represented by scores ≥4 [19].

## Statistical analyses

Descriptive statistics included mean, median, and standard deviation. The median of gene expression was the cut-off value used to categorize into high or low expression. Whereas the protein expression patterns were determined by scores ≤4 or >4. Associations of protein expression according to categorical clinicopathological data were assessed through Chi-square or Fisher's tests as appropriate. Immunolabeling scores were analyzed using the non-parametric tests Mann-Whitney and Kruskal-Wallis. Overall survival (OS) time was calculated from the date of surgery to the date of death or last follow-up. Kaplan-Meier curve and the log-rank test were applied to evaluate differences regarding MTHFD2 expression profiles and OS. All analyses were performed using IBM® SPSS® v.20, with significance level set at α = 0.05.

## Results

### *MTHFD2* gene expression predicts RCC prognosis

The prognostic potential of the *MTHFD2* was assessed by mRNA expression using two Km-plotter platform databases. We observed that tumor samples exhibited significantly higher *MTHFD2* than NN tissues. Importantly, metastatic samples presented even higher expression compared to tumor samples alone (Fig 1A, p = 6.64e-69). Comparing the RCC histologic subtypes to NN kidney samples, ccRCC showed higher gene expression than NN tissues, the pRCC subtype exhibited the opposite profile and chRCC was quite similar to NN (Fig 1B–1D). It is worth noting that *MTHFD2* higher expression was significantly associated with shorter overall survival for the ccRCC and pRCC subtypes (Fig 2A and 2C). Interestingly, no associations with recurrence-free survival were found for ccRCC (Fig 2B), whereas *MTHFD2* overexpression suggested shorter RFS in the pRCC cohort (Fig 2D).

### MTHFD2 immunostaining revels RCC aggressiveness

Considering the relevance of *MTHFD2* gene expression in RCC prognosis, we investigated its protein profile in a cohort with 117 RCC. The immunostaining pattern was predominantly cytoplasmic with membranous positivity found in a few cases only. Regarding the percentage of cells, 66 out of 117 (56.4%) cases presented more than 76% of positive cells, 19 (16.2%) had <5%, four cases (3.4%) had 6%-25%, 10 (8.5%) had 26–50%, and 18 cases (15.4%) showed 51% -75% of stained cells. Nineteen tumors were negative (16.2%), 68 presented weak intensity

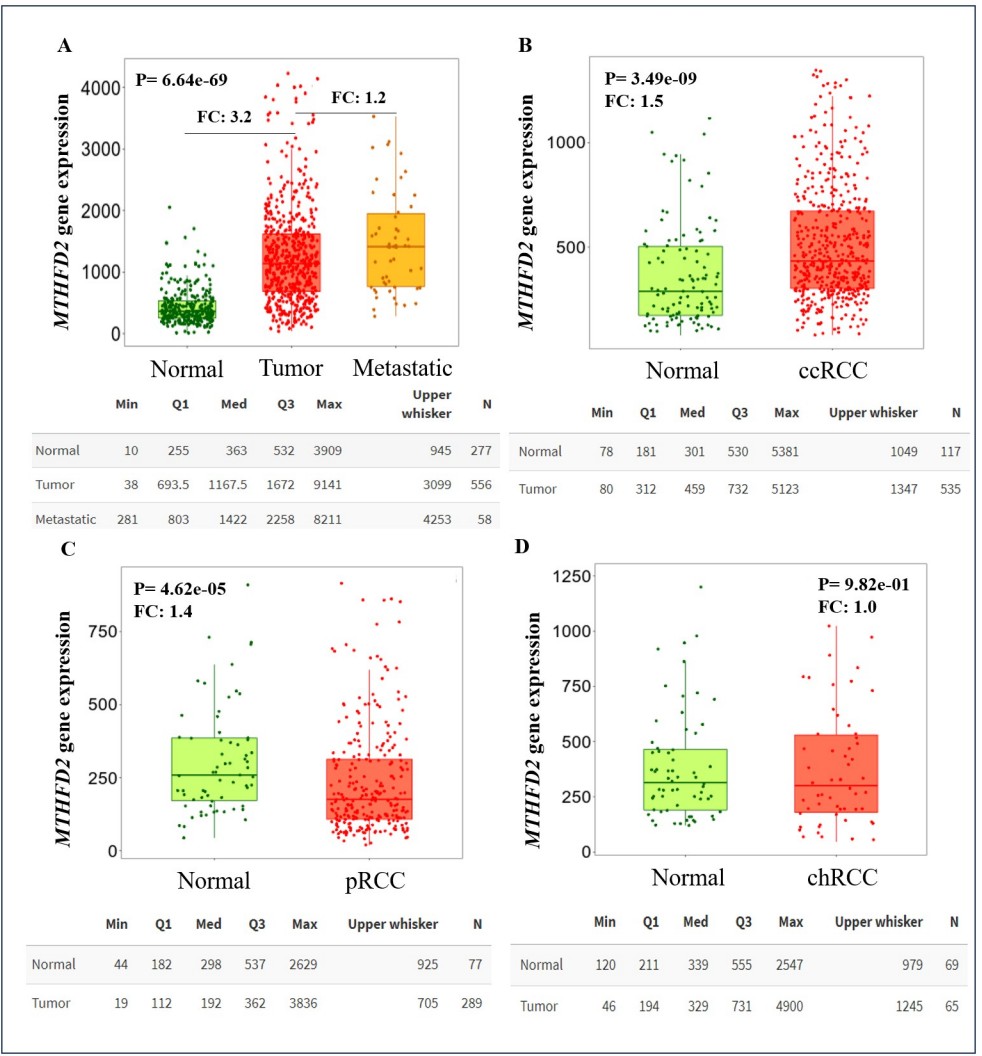

**Fig 1. *MTHFD2* gene expression in renal cell carcinoma.** (A) Comparison between tumor and non-neoplastic samples analyzed from the Gene chip database. (B-D) Expression in ccRCC, pRCC and chRCC histological subtypes compared to non-neoplastic specimens investigated in the Pan-cancer database.

(58.1%), 21 were moderate (17.9%) and nine (7.7%) had strong staining (Fig 3A–3C). The immunohistochemistry score revealed high MTHFD2 expression (score ≥4) in 60.7% of cases regardless of the histological subtype.

Interestingly, MTHFD2 expression varied according to the RCC subtypes. The intensity (Fig 3D), as well as the immunolabeling scores (Fig 3E), was significantly lower in ccRCC compared to pRCC and ccRCC (p<0.05). The majority of pRCC (20 out of 21) and all five chRCC cases exhibited high expression (scores ≥4), while the proportion of ccRCC cases within low and high groups was quite similar (p<0.001), Table 1.

We found significant associations between MTHFD2 protein expression, the tumor Fuhrman grade, and diameter, Table 1. In tumors classified as high expression (scores ≥4), 39/71 (54.9%) cases were larger than 7 cm, while only 16/45 (35.6%) cases with low expression were larger tumors (Odds ratio = 2.2; 95%CI: 1.024–4.765; p = 0.042). Similarly, 37 out of 71 cases (52.1%) with high MTHFD2 were Fuhrman grades 3 and 4, while in the low expression group,

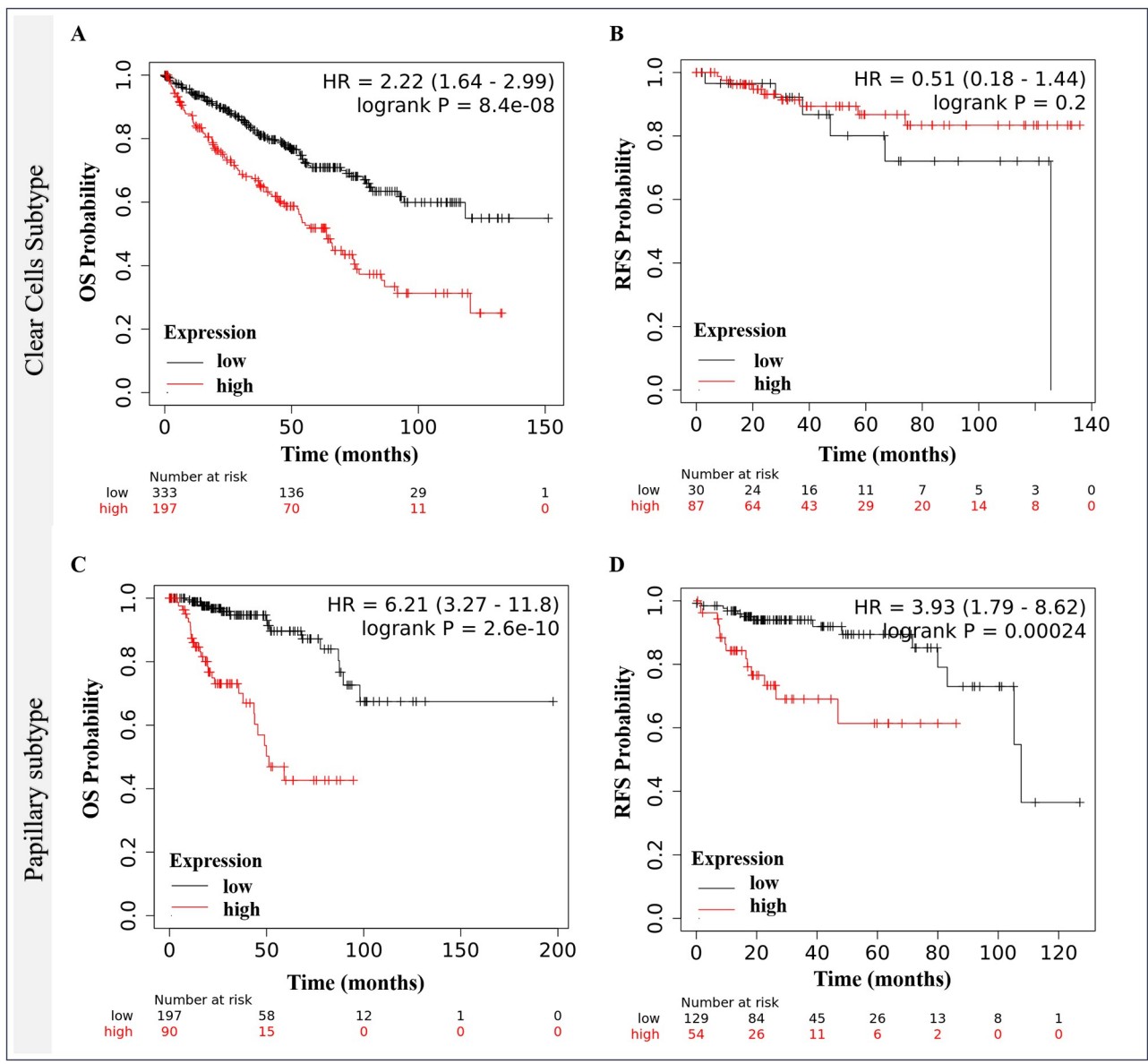

**Fig 2. Associations of *MTHFD2* gene pattern with patients' survival.** (A) and (C) Overall survivals in the ccRCC and pRCC cohorts. (B) and (D) Recurrence-free survivals in patients with ccRCC and pRCC. Kaplan-Meier curves and log-rank test. Pan-cancer database.

few cases 11/47 (23.4%) had advanced Fuhrman grades (Odds ratio = 3.918; 95%CI: 1.689–9.086; p = 0.001). Also, the immunolabeling score increased according to Fuhrman grades, being higher in tumors 3 and 4 compared to 1 and 2 (Fig 4A, p = 0.003). Specifically, the scores were significantly different between grades 1 and 3 (Fig 4B, p = 0.020). Considering the tumor subtypes separately, the ccRCC maintained a significant association between MTHFD2 expression and Fuhrman grade (Odds ratio = 4.364; 95%CI: 1.719–11.079; p = 0.001). In this group, the score was higher in Fuhrman grades 3 and 4 compared to grades 1 and 2 (Fig 4C, p = 0.009). Significant difference was specifically found between grades 1 and 3 (Fig 4D, p = 0.011), which was not observed for pRCC and chRCC cases. Additionally, we identified

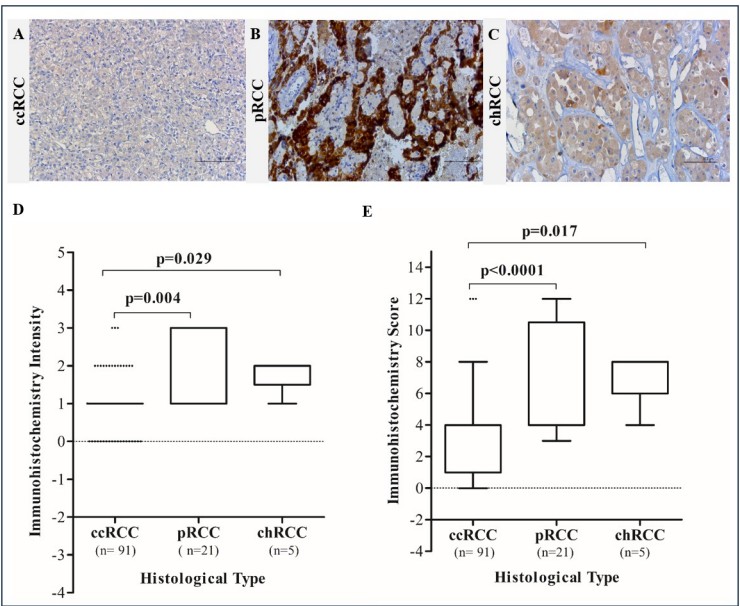

**Fig 3. MTHFD2 protein expression in renal cell carcinoma by immunohistochemistry.** (A) Representative image of ccRCC case with weak intensity. (B) Representative image of pRCC case with strong intensity. (C) chRCC case with moderate intensity. (D) Intensity immunolabeling variation between RCC subtypes. (E) Comparison of expression scores between RCC subtypes. Kruskal-Wallis test.

higher proportion of samples with elevated MTHFD2 protein expression in tumors with necrosis (63.4%), however without statistical significance (p = 0.097).

The prognostic value of MTHFD2 protein expression was not confirmed through OS analysis. However, patients classified as high expression had shorter survival rates (S1 Fig). No other significant association was observed between the protein expression and clinical-epidemiological data, such as gender, age, race, presence of symptoms (haematuria, lumbar pain and palpable mass), comorbidities (diabetes, renal hypertension and kidney disease), Body Mass Index (BMI), alcohol and tobacco consumption, family history of cancer, nor with histopathological data, including lymph node involvement, metastases, angiolymphatic invasion and laterality.

## Discussion

The renal cell carcinoma is recognized by an extensive energy metabolism reprogramming, which is related to mutations in specific genes, including *VHL*, *MET*, *BAP1*, *TFE3*, *TFEB*, *FLCN*, *MITF*, *FH*, *SDHB*, *SDHC*, *SDHD*, *TSC1*, *TSC2*, *PBRM1*, *SETD2* and *KDM5C*. In general, these genes are closely interconnected to various metabolic pathways, contributing to the complexity and uniqueness of this cancer [12, 27–29]. The "Warburg Effect" is the main metabolic change observed, where cancer cells restrict the oxidative phosphorylation and oxidation of fatty acids in the mitochondria to use the aerobic glycolysis to meet their energy demands. It places lactate as the crucial by-product, being directly linked to a favorable microenvironment for tumor growth, angiogenesis, and therapy resistance [12, 30–32]. In addition, the adverse impact on lipid metabolism leads to unoxidized lipids accumulation, contributing to the RCC metabolic complexity [10, 30, 33, 34]. In this context, it has become evident that MTHFD2 plays a key role as a significant regulator in the mitochondrial folate pathway in

**Table 1.** MTHFD2 expression according to the clinical and pathological features.

| Clinicopathological feature | N = 117 | MTHFD2 expression | | P value |
|---|---|---|---|---|
| | | High n (%) | Low n (%) | |
| **Sex** | | | | n.s. |
| Male | 77 | 47 (66.2%) | 30 (65.2%) | |
| Female | 40 | 24 (33.8%) | 16 (34.8%) | |
| **Age** | | | | n.s. |
| ≥50 Years old | 101 | 61 (85.9%) | 40 (87%) | |
| <50 Years old | 16 | 6 (13%) | 10 (14.1%) | |
| **Symptoms*** | | | | n.s. |
| No | 89 | 53 (77.9%) | 36 (85.7%) | |
| Yes | 21 | 15 (22.1%) | 6 (14.3%) | |
| **Smoking and/or alcohol*** | | | | n.s. |
| No | 42 | 23 (37.7%) | 19 (45.2%) | |
| Yes | 61 | 38 (62.3%) | 23 (54.8%) | |
| **Laterality*** | | | | n.s. |
| Left kidney | 52 | 32 (45.7%) | 20 (44.4%) | |
| Right Kidney | 63 | 38 (54.3%) | 25 (55.6%) | |
| **Histopatholgical subtype** | | | | **<0.0001** |
| ccRCC | 91 | 46 (64.8%) | 45 (97.8%) | |
| pRCC | 21 | 20 (28.2%) | 1 (2.2%) | |
| chRCC | 5 | 5 (7%) | 0 (0%) | |
| **Tumor necrosis** | | | | 0.097 |
| No | 50 | 26 (36.6%) | 24 (52.2%) | |
| Yes | 67 | 45 (63.4%) | 22 (47.8%) | |
| **Fuhrman grade** | | | | *0.001* |
| 1 and 2 | 70 | 34 (47.9%) | 36 (78.3%) | |
| 3 and 4 | 47 | 37 (52.1%) | 10 (21.7%) | |
| **Tumor diameter*** | | | | *0.042* |
| < 7cm | 61 | 32 (45.1%) | 29 (64.4%) | |
| > 7 cm | 55 | 39 (54.9%) | 16 (35.6%) | |
| **Staging** | | | | n.s. |
| I and II | 77 | 47 (66.2%) | 30 (65.2%) | |
| III and IV | 40 | 24 (33.8%) | 16 (34.8%) | |
| **Metastasis*** | | | | n.s. |
| No | 104 | 62 (89.9%) | 42 (93.3%) | |
| Yes | 10 | 7 (10.1%) | 3 (6.7%) | |
| **Surgical intervention*** | | | | n.s. |
| Radical | 92 | 55 (77.5%) | 37 (84.1%) | |
| Partial | 23 | 16 (22.5%) | 8 (15.9%) | |
| **Death*** | | | | n.s. |
| No | 98 | 58 (84.1%) | 40 (88.9%) | |
| Yes | 16 | 11 (15.9%) | 5 (11.1%) | |

*Complete data not available. Chi-square or Fisher's test. n.s.: not significant.

cancer cell metabolism. Emerging as a potential prognostic biomarker in various types of cancer [22]. However, no specific parameters have been stablished in order to apply the gene or protein investigation in the RCC diagnosis. This study evaluated MTHFD2 relationship with clinical and pathological features of RCC cohorts, by gene and *in situ* expression methods.

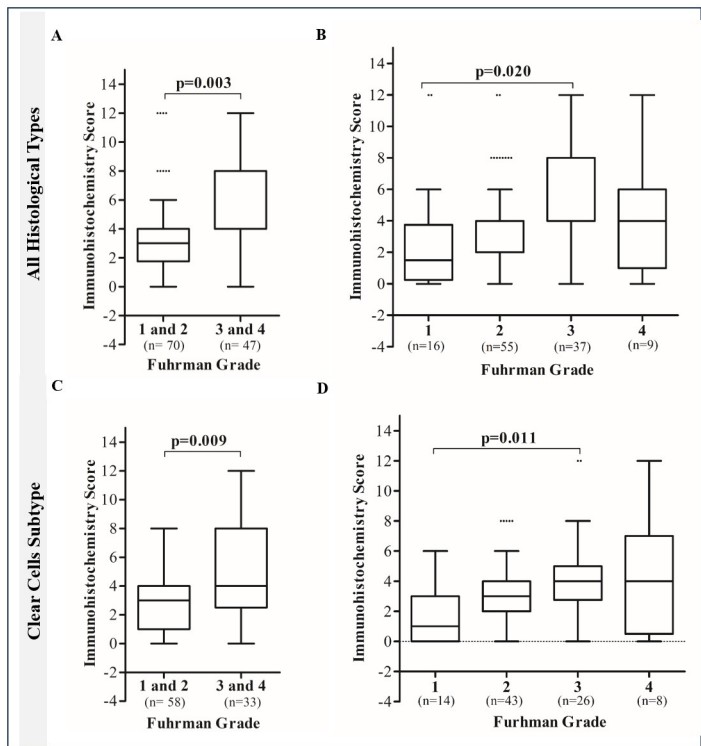

**Fig 4. MTHFD2 protein expression scores according to renal cell carcinoma Fuhrman grades.** (A) Increased immunolabeling score in tumors with Fuhrman grades 3 and 4, considering all histological subtypes. (B) Significant differences between grades 1 and 3, considering all histological subtypes. (C) Scores according to Fuhrman grades 1 and 2 *versus* 3 and 4 in the clear cell subtype. (D) Scores according to Fuhrman grades separately.

Studies on TCGA databases highlighted the *MTHFD2* overexpression in tumor samples compared to NN ones. A previous study [35] on different tumors observed higher *MTHFD2* profile in 25 out of 31 types of cancer (80.6%), including the ccRCC subtype. Our findings revealed that, compared to NN samples, the ccRCC presents elevated *MTHFD2* expression. In contrast, the pRCC exhibits significantly lower levels and chRCC subtype showed similar expression pattern. These corroborate with Green et al., 2019 findings, that evaluated the gene expression in a ccRCC cohort and NN samples [36]. As expected for an oncogene candidate, *MTHFD2* levels was found increased in other types of cancer, such as colorectal cancer [35], urothelial carcinoma [37], breast cancer [38], esophageal squamous cell carcinoma [39], as well as lung adenocarcinoma [40].

Considering MTHFD2 contribution to the redox homeostasis through NADPH production, its elevated levels promote suitable environment for cancer progression and metastasis [22]. The enzyme supports the nucleotide synthesis, which is essential to sustain high tumor cell proliferation rates [41]. In addition to the differences between RCC and normal kidney, we observed *MTHFD2* gene expression even higher in RCC metastatic samples, suggesting it promotes an aggressive behavior. In addition, *MTHFD2* was associated with overall survival in patients with ccRCC and pRCC, and with recurrence-free survival in pRCC group. These findings agree with studies showing *MTHFD2* gene expression as a prognostic biomarker in breast cancer [38, 42], colorectal cancer [35], and urothelial carcinoma [37]. Concerning to RCC, a report on metabolism-related biomarkers in ccRCC and pRCC revealed seven high risk genes.

Among them, *MTHFD2* overexpression was associated with poor prognosis and patients" death [43]. Consistently and in support of our findings, Green et al. 2019 observed overexpression of *MTHFD2* gene associated with tumor stages II, III and IV, as well as shorter survival rates in a ccRCC group. They were also associated with higher stages (II, III and IV) [36].

To validate the impressive results of *MTHFD2* transcript, we investigated the protein pattern in 117 RCC tissues through immunohistochemistry, which is a low-cost method used to complement pathological diagnosis of cancer [44, 45]. Despite the accessible and simple results provided by immunolabeling, few reports have explored *in situ* MTHFD2 expression to evaluate tumor prognosis [19, 22, 46–49]. In our cohort, the high expression was observed in 60.7% of cases, regardless of histological subtype. This profile was more frequent and with more elevated scores in pRCC and chRCC tumors than in ccRCC. Lin et al. 2018 examined protein expression of MTHFD2 in 137 RCC tissues and identified high expression in most of cases (58.4%), especially in the clear cell subtype [19]. It is important to mention that Lin et al., 2018 have published an erratum regarding authors' contribution [50], which does not modify their findings. MTHFD2 increased pattern was also observed in a small cohort of ccRCC (33 cases) compared to normal tissues (34 cases), although the study did not establish associations with clinical, histopathological or survival data [36]. Furthermore, by *in vivo* and *in vitro* experiments, *MTHFD2* knockdown reduced tumor size and impaired cell proliferation, migration, and invasion [36]. These findings highlight the essential role of MTHFD2 in RCC tumor progression. Moreover, *MTHFD2* positively regulates HIF-2α, favoring the glycolytic activity in ccRCC, which is known to be associated with hypoxia-inducible factor (HIF) elevated levels. In the specific context of ccRCC, more than 80% of cases present HIF-α accumulation and transcript amplification of HIF target genes due to loss of the von Hippel-Lindau (VHL) function [36]. However, for the other RCC subtypes, the MTHFD2 may be an important trigger of metabolic disorders, contributing to an aggressive tumor behavior.

Regarding aggressiveness, we identified predominant MTHFD2 expression in larger tumors ($> 7cm$), which means T2, -T3, and -T4 according to TNM, AJCC 8th. Although no significant association was found according to RCC tumoral stages, our result is in line with previous studies with RCC and esophageal squamous cell carcinoma, that reported elevated protein expression associated with aggressive AJCC tumor stages, considering I/II *versus* III/IV [19, 39]. In bladder cancer, the gene expression suggests advanced tumor stages (according to AJCC stages), while the protein pattern showed a similar trend [49]. Furthermore, we observed a significant association between higher MTHFD2 expression and higher Fuhrman grade, which was also seen when evaluating the ccRCC separately. To date, no studies have analyzed this parameter according to the MTHFD2 expression profile. It is worth noting that Fuhrman grade is a tumor classification based on the nuclear morphology and is considered a prognostic factor by AJCC 8th edition guideline [18].

Interestingly, we identified a higher proportion of samples with elevated MTHFD2 protein expression in tumors with necrosis, which did not reach statistical significance. According to the 2017 AJCC guideline, the presence of necrosis is a prognostic factor for RCC. However, studies on the MTHFD2 gene or protein expression in RCC have not assessed this specific parameter [19, 38]. In our analysis, the MTHFD2 protein pattern was not associated with OS and RFS in patients with RCC, and its overexpression only suggested shorter overall survival in the group with pRCC, without statistical significance. Three different studies described significant associations with patients' survival by assessing MTHFD2 immunolabeling in 78 esophageal carcinoma samples, 323 lung cancer samples, and 137 RCC cases [19, 39, 47, 48]. It is important to note that most of our RCC cases has a favorable outcome, and that 16 of 117 patients died. Thus, results should be interpreted with caution due to the limited number of cases in the individual groups.

## Conclusions

In summary, our results strongly point to the relevant role of MTHFD2 expression in RCC, with significant implications for the prognosis and tumor aggressiveness. The gene overexpression is associated with unfavorable outcome such as metastasis and shorter OS in ccRCC and pRCC subtypes, as well as shorter RFS in pRCC patients. These emphasize the MTHFD2 prognostic value, in a general context of RCC, whereas underline the need to explore less frequently addressed RCC subtypes, such as papillary. Through immunolabeling, MTHFD2 high expression was associated with higher Fuhrman grade and larger tumors. The variation of MTHFD2 patterns found between the different histological subtypes highlights the disease heterogeneity. Nonetheless, the protein expression did not confirm the prognosis, being observed a trend in the pRCC subtype.

These results raise questions about MTHFD2 gene expression as a valuable prognostic biomarker. Also, emphasize the importance of further studies with larger and subtype specific cohorts, to validate the significance of *in situ* investigation for the RCC prognosis and clinical management.

## Supporting information

**S1 Table. Patients' clinical, epidemiological and histopathological data.**
(DOCX)

**S1 Fig. Overall survival analysis according to MTHFD2 protein expression in RCC and subtypes.** (A) RCC survival curve, considering all subtypes. (B) For ccRCC. (C) For pRCC. (D) For chRCC.
(TIF)

**S1 Appendix. Spreadsheet with all the data from the cases included in the survey.**
(XLSX)

## Acknowledgments

We thank the Centro de Pesquisa em Rim–CePRim of Federal University of Triângulo Mineiro–UFTM for the academic and for performing the immunohistochemistry technic. We thank the Department of Clinical Surgery of Federal University of Triângulo Mineiro–UFTM for handling the tissue samples.

## Author Contributions

**Conceptualization:** Régia C. P. Lira.

**Data curation:** Rafaela V. N. Silva, Lucas A. Berzotti, Marcella G. Laia, Millena Brandão.

**Formal analysis:** Rafaela V. N. Silva, Lucas A. Berzotti, Régia C. P. Lira.

**Investigation:** Rafaela V. N. Silva, Lucas A. Berzotti, Marcella G. Laia, Liliane S. Araújo, Crislaine A. Silva, Karen B. Ribeiro, Adilha M. R. Michelleti.

**Methodology:** Rafaela V. N. Silva, Liliane S. Araújo, Régia C. P. Lira.

**Supervision:** Régia C. P. Lira.

**Writing – original draft:** Rafaela V. N. Silva, Régia C. P. Lira.

**Writing – review & editing:** Rafaela V. N. Silva, Karen B. Ribeiro, Millena Brandão, Adilha M. R. Michelleti, Juliana R. Machado, Régia C. P. Lira.

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
