## [Decision Letter · Decision Letter 0]

8 Jan 2024

PONE-D-23-42274Implications of MTHFD2 expression in renal cell carcinoma aggressivenessPLOS ONE

Dear Dr. Lira,

Thank you for submitting your manuscript to PLOS ONE. After careful consideration, we feel that it has merit but does not fully meet PLOS ONE’s publication criteria as it currently stands. Therefore, we invite you to submit a revised version of the manuscript that addresses the points raised during the review process.

We look forward to receiving your revised manuscript.

Kind regards,

Giuseppe Lucarelli, M.D., Ph.D.

Academic Editor

PLOS ONE

Journal Requirements:

2. We note that your Data Availability Statement is currently as follows: "All relevant data are within the manuscript and its Supporting Information files."

4. Please upload a copy of Supporting Information Figure/Table/etc. Table S1 which you refer to in your text on page 15.

Reviewers' comments:

Reviewer's Responses to Questions

**Comments to the Author**

1. Is the manuscript technically sound, and do the data support the conclusions?

Reviewer #1: Yes

Reviewer #2: Yes

2. Has the statistical analysis been performed appropriately and rigorously? 

Reviewer #1: Yes

Reviewer #2: Yes

3. Have the authors made all data underlying the findings in their manuscript fully available?

Reviewer #1: Yes

Reviewer #2: Yes

4. Is the manuscript presented in an intelligible fashion and written in standard English?

Reviewer #1: Yes

Reviewer #2: Yes

5. Review Comments to the Author

Reviewer #1: In this study the authors investigated the prognostic role of MTHFD2 in different RCC cohorts.

I have some comments:

-Renal cell carcinoma is essentially a metabolic disease characterized by a reprogramming of energetic metabolism (PMID: 36960789; PMID: 30983433, PMID: 36430837,PMID: 36310399). In particular the metabolic flux through glycolysis is partitioned (PMID: 29371925, PMID: 28933387, PMID: 25945836), and mitochondrial bioenergetics and OxPhox are impaired , as well as lipid metabolism (PMID: 30538212; PMID: 32861643, PMID: 29371925, PMID: 36430448). In this scenario it has been shown that MTHFD2 is an important regulator of mitochondrial folate pathway in cancer cell metabolism. These findings should be referenced and discussed.

Reviewer #2: This study evaluated the role of MTHFD2 in RCC associating its expression with tumor characteristics and prognostic factors.

I have some comments:

renal cell carcinoma is one of the most immune-infiltrated tumors (PMID: 31527133, PMID: 30738745; PMID: 27063186). Emerging evidence suggests that the activation of specific metabolic pathway have a role in regulating angiogenesis and inflammatory signatures (PMID: 32345771, PMID: 28359744). Features of the tumor microenvironment heavily affect disease biology and may affect responses to systemic therapy (PMID: 38003705; PMID: 37189689; PMID: 33265926; PMID: 36902242; PMID: 37373581).MTHFD2 is a metabolic checkpoint controlling effector and regulatory T cell fate and function (PMID: 34767747) and can modulate immune cell infiltration and regulate immunoflogosis. These processes should be explored and discussed.

6. PLOS authors have the option to publish the peer review history of their article (what does this mean?). If published, this will include your full peer review and any attached files.

Reviewer #1: No

Reviewer #2: No

---

## [Author Response · Author response to Decision Letter 0]

7 Feb 2024

RE: PONE-D-23-42274: Implications of MTHFD2 expression in renal cell carcinoma aggressiveness 

Dear Emily Chenette, PhD   

Editor-in-Chief, PLOS ONE   

We would like to express our gratitude for the care given to our manuscript mentioned above. In this revised version, we have taken on board the observations made by you and the reviewers, making substantial changes as suggested. We are confident that the changes implemented have contributed significantly to improving the quality and content of the revised manuscript. We hope that these additions will prove beneficial in strengthening the arguments presented in our article. 

Specifically, we are including: 

The file format of the manuscript has been adjusted to meet the style requirements of PLOS ONE, including adjusting the nomenclature of the files. 

We incorporated an Excel spreadsheet containing all the data of the patients involved in the research, ensuring that only identification was kept anonymous, in compliance with the guidelines of the ethics committee. 

We add that all the authors are unanimous on the importance of sharing data prior to acceptance, and we confirm our consent. 

S1 Table has been uploaded. 

The erratum of one article was included in the references and in the discussion section. 

We incorporated the reviewers' suggestions to improve our manuscript. 

As suggested, the figures have been uploaded to the PACE platform (https://pacev2.apexcovantage.com/) to ensure the quality requirements. 

We would like to thank you for your interest and the opportunity to revise our manuscript and we hope to have satisfactorily addressed the points raised by the reviewers. 

Additionally, we would like to modify our financial disclosure, once our institution does not have resources for publishing payments and the research grant provides no support for Article Publishing Charges. Thus, we respectfully request a waiver or discount on the publication fee. This concession will contribute significantly to disseminate our research and to promote the inclusion of independent researchers in the academic scene. 

With our best regards, and on behalf of the co-authors, 

Régia Lira, PhD 

Rebuttal letter 

Dear reviewers, thank you very much for your constructive comments and valuable suggestions regarding our article. Your expertise was fundamental in improving the quality of the study. We have taken into account the observations made, making substantial changes as recommended. We believe that these changes have contributed significantly to improving the manuscript. Follow the changes: 

Journal Requirements: 

1- Please ensure that your manuscript meets PLOS ONE's style requirements, including those for file naming. The PLOS ONE style templates can be found at  

https://journals.plos.org/plosone/s/file?id=wjVg/PLOSOne_formatting_sample_main_body.pdf and 

RE: The manuscript file format was adjusted to fully comply with the requirements established by PLOS ONE, specifically in terms of file nomenclature. This adaptation aims to ensure strict adherence to the journal's editorial guidelines, contributing to the uniformity and standardization of the documents submitted. The changes are highlighted in the text. 

2- We note that your Data Availability Statement is currently as follows: "All relevant data are within the manuscript and its Supporting Information files." 

If your submission does not contain these data, please either upload them as Supporting Information files or deposit them to a stable, public repository and provide us with the relevant URLs, DOIs, or accession numbers. For a list of recommended repositories, please see https://journals.plos.org/plosone/s/recommended-repositories. 

RE: We really appreciated this input. For this reason, we included an Excel spreadsheet with all data analyzed in our study, which is available as supporting information (S1 Appendix). We would like to point it out that information regarding participants identity was replaced by random numbers, taking care to preserve their anonymity in accordance with ethics committee guidelines. This initiative aims to promote the transparency and replicability of the study, allowing access the detailed information underlying the research, which reiterates our commitment to scientific integrity and open data sharing. 

3-   When completing the data availability statement of the submission form, you indicated that you will make your data available on acceptance. We strongly recommend all authors decide on a data sharing plan before acceptance, as the process can be lengthy and hold up publication timelines. Please note that, though access restrictions are acceptable now, your entire data will need to be made freely accessible if your manuscript is accepted for publication. This policy applies to all data except where public deposition would breach compliance with the protocol approved by your research ethics board. If you are unable to adhere to our open data policy, please kindly revise your statement to explain your reasoning and we will seek the editor's input on an exemption. Please be assured that, once you have provided your new statement, the assessment of your exemption will not hold up the peer review process. 

RE: We unanimously ratified our consent to the open data policy, recognizing that this practice contributes to transparency and the advancement of scientific knowledge. Thus, our entire data was summarized in the S1 Appendix, which is available as supporting information. 

4- Please upload a copy of Supporting Information Figure/Table/etc. Table S1 which you refer to in your text on page 15. 

RE: We apologize for the inconvenience, and the Table S1 was uploaded as supporting information. 

5- Please review your reference list to ensure that it is complete and correct. If you have cited papers that have been retracted, please include the rationale for doing so in the manuscript text or remove these references and replace them with relevant current references. Any changes to the reference list should be mentioned in the rebuttal letter that accompanies your revised manuscript. If you need to cite a retracted article, indicate the article’s retracted status in the References list and also include a citation and full reference for the retraction notice. 

RE: We appreciate the Editor raising this point. After a detailed review of our references, we decided to keep the Lin et al., 2018 reference, considering that we have identified only two studies addressing MTHFD2 protein expression in Renal cell carcinoma (Lin et al., 2018 - DOI: 10.1159/000495402, Green et al., 2019 - DOI: 10.1038/s41388-019-0869-4) through immunohistochemistry, which are relevant to the discussion and understanding of our findings. Furthermore, it is important to mention that the erratum published by Lin et al., 2018 (DOI: 10.33594/000000114.) clarifies a misunderstanding regarding the authors' contribution and does not concern the study reliability. We have followed your suggestion and indicated the article’s retracted status in the References list as well as its full reference [51]. In addition, we have included the retraction citation in the discussion section and have added the sentence: 

 “…It is important to mention that Lin et al., 2018 have published an erratum regarding authors' contribution [50], which does not modify their findings.” (line: 300) 

Reviewer #1 (Remarks to the Author): Some improvements suggested: 

In this study the authors investigated the prognostic role of MTHFD2 in different RCC cohorts. 

I have some comments: Renal cell carcinoma is essentially a metabolic disease characterized by a reprogramming of energetic metabolism (PMID: 36960789; PMID: 30983433, PMID: 36430837, PMID: 36310399). In particular the metabolic flux through glycolysis is partitioned (PMID: 29371925, PMID: 28933387, PMID: 25945836), and mitochondrial bioenergetics and OxPhox are impaired, as well as lipid metabolism (PMID: 30538212; PMID: 32861643, PMID: 29371925, PMID: 36430448). In this scenario it has been shown that MTHFD2 is an important regulator of mitochondrial folate pathway in cancer cell metabolism. These findings should be referenced and discussed. 

RE: I would like to sincerely thank you for your valuable comments. We enthusiastically adhered to your suggestions by adding in the discussion section the sentence: 

“…The renal cell carcinoma is recognized by an extensive energy metabolism reprogramming, which is related to mutations in specific genes, including VHL, MET, BAP1, TFE3, TFEB, FLCN, MITF, FH, SDHB, SDHC, SDHD, TSC1, TSC2, PBRM1, SETD2 and KDM5C. In general, these genes are closely interconnected to various metabolic pathways, contributing to the complexity and uniqueness of this cancer [12,27–29]. The "Warburg Effect" is the main metabolic change observed, where cancer cells restrict the oxidative phosphorylation and oxidation of fatty acids in the mitochondria to use the aerobic glycolysis to meet their energy demands. It places lactate as the crucial by-product, being directly linked to a favorable microenvironment for tumor growth, angiogenesis, and therapy resistance [12,30–32]. In addition, the adverse impact on lipid metabolism leads to unoxidized lipids accumulation, contributing to the RCC metabolic complexity [10,30,33,34].” (line: 250) 

We are confident that the modifications implemented have improved the quality of our discussion. 

Reviewer #2 (Remarks to the Author): Some improvements suggested: 

This study evaluated the role of MTHFD2 in RCC associating its expression with tumor characteristics and prognostic factors. 

I have some comments: renal cell carcinoma is one of the most immune-infiltrated tumors (PMID: 31527133, PMID: 30738745; PMID: 27063186). Emerging evidence suggests that the activation of specific metabolic pathway have a role in regulating angiogenesis and inflammatory signatures (PMID: 32345771, PMID: 28359744). Features of the tumor microenvironment heavily affect disease biology and may affect responses to systemic therapy (PMID: 38003705; PMID: 37189689; PMID: 33265926; PMID: 36902242; PMID: 37373581). MTHFD2 is a metabolic checkpoint controlling effector and regulatory T cell fate and function (PMID: 34767747) and can modulate immune cell infiltration and regulate immunoflogosis. These processes should be explored and discussed. 

RE: We would like to express our gratitude for the constructive suggestions. Your observations have been carefully incorporated in the introduction heading in two sentences: 

“... In addition to the specific genetic profiles, such as genes mutations, the tumor microenvironment exerts influence on the tumor biology, playing a crucial role in the therapy response [7–12]. The RCC stands out for having one of the greatest immunological infiltrations compared to other types of cancer, highlighting its aggressive nature. The Warburg effect and activation of specific metabolic pathways are known to promote angiogenesis, inflammatory signatures and antioxidant defense, which are associated with impairment of chemotherapy and radiotherapy and malignant behavior [13–16]”. (line: 56) 

And 

“… Furthermore, MTHFD2 stands out as a crucial metabolic checkpoint, that regulates both the effector and regulatory T cells, suggesting a broader role for this enzyme in RCC microenvironment context [23].” (line: 78) 

Since our analysis have not explored the MTHFD2 regulatory functions in the immunological field, we considered the sentences would be valuable for the introduction section instead of the discussion one. We thank you for guiding us in the pursuit of excellence. 

Additional modifications: 

We have excluded the headings Introduction, Methods, Results and Conclusion from the abstract text in accordance with PLOS ONE's style. 

We have removed the authors' contributions description from the front page to suit the PLOS ONE's style, since it was already included in the submission form. The competition of interest and funding headings were also removed from the manuscript text for the same reason.

---

## [Editor Report · Decision Letter 1]

9 Feb 2024

Implications of MTHFD2 expression in renal cell carcinoma aggressiveness

PONE-D-23-42274R1

Dear Dr. Lira,

We’re pleased to inform you that your manuscript has been judged scientifically suitable for publication and will be formally accepted for publication once it meets all outstanding technical requirements.

Kind regards,

Giuseppe Lucarelli, M.D., Ph.D.

Academic Editor

PLOS ONE
---

## [Editor Report · Acceptance letter]

21 Feb 2024

PONE-D-23-42274R1 

PLOS ONE

Dear Dr. Lira, 

I'm pleased to inform you that your manuscript has been deemed suitable for publication in PLOS ONE. Congratulations! Your manuscript is now being handed over to our production team.

Kind regards, 

on behalf of

Dr. Giuseppe Lucarelli 

Academic Editor

PLOS ONE